# Research Progress on Glycoprotein 5 of Porcine Reproductive and Respiratory Syndrome Virus

**DOI:** 10.3390/ani13050813

**Published:** 2023-02-23

**Authors:** Qin Luo, Yajie Zheng, Hang Zhang, Zhiyu Yang, Huiyang Sha, Weili Kong, Mengmeng Zhao, Nina Wang

**Affiliations:** 1School of Life Science and Engineering, Foshan University, Foshan 528231, China; 2Gladstone Institutes of Virology and Immunology, University of California, San Francisco, CA 94158, USA

**Keywords:** porcine reproductive and respiratory syndrome virus, GP5 protein, host proteins, vaccines

## Abstract

**Simple Summary:**

Research on GP5 protein is of great significance in the diagnosis, prevention, and control of porcine reproductive and respiratory syndrome virus (PRRSV). We summarize its genetic variation, immunity, replication, apoptosis, virulence, interaction with viral protein and host proteins, which provides a theoretical foundation for exploring the PRRSV replication mechanisms and developing new vaccines.

**Abstract:**

Porcine reproductive and respiratory syndrome (PRRS) is an acute, febrile, and highly contagious disease caused by the porcine reproductive and respiratory syndrome virus (PRRSV). Glycoprotein 5 (GP5) is a glycosylated envelope protein encoded by the PRRSV *ORF5*, which has good immunogenicity and can induce the body to produce neutralizing antibodies. Therefore, study of GP5 protein is of great significance in the diagnosis, prevention, and control of PRRSV and the development of new vaccines. We reviewed GP5 protein genetic variation, immune function, interaction with viral protein and host proteins, induction of cell apoptosis, and stimulation of neutralizing antibodies. GP5 protein’s influence on virus replication and virulence, as well as its use as a target for viral detection and immunization are reviewed.

## 1. Introduction

Porcine reproductive and respiratory syndrome (PRRS) is a severe infectious disease caused by the porcine reproductive and respiratory syndrome virus (PRRSV). The disease is characterized by adverse reproductive outcomes such as miscarriage, premature delivery, stillbirth, and fetal mummification. Non-reproductive effects are exhibited as respiratory diseases, immunosuppression, secondary diseases, and increased piglet mortality. These effects combine to cause massive economic losses to the swine industry worldwide.

PRRS outbreaks initially occurred in the United States in 1987, and then spread to Europe and Asia [1]; it was initially called the “mystery pig disease” by the American Animal Health Society in 1990 [2]. In 1991, PRRSV was isolated from diseased animals in the Netherlands and named Lelystad virus (LV) [3]. PRRSV was isolated in the United States in 1992 and named SIRS virus (also known as VR-2332) [4]. According to its genetic and antigenic characteristics, PRRSV can be divided into two genotypes: the European type (PRRSV-1) represented by the LV strain and the American type (PRRSV-2) represented by the VR-2332 strain [5]. The two types share approximately 60% of their nucleotide identity [6]. The first PRRSV strain of China, isolated and identified from aborted fetuses in 1996, was classified as PRRSV-2 [7]. In 2006, an outbreak of highly pathogenic PRRSV (HP-PRRSV) characterized by high mortality, fever, and abortion rates devastated the Chinese swine industry [8].

PRRSV is an RNA virus with an envelope and single positive strand [9], belonging to the order *Nidovirales* and the family *Arteriviridae.* It also includes lactate dehydrogenase-elevating virus of mice, equine arteritis virus, and simian hemorrhagic fever virus [10]. The total length of the genome is approximately 15 kb, which forms a cap structure at the 5′ end during mRNA processing, and has a poly-A tail structure at the 3′ end [11]. The PRRSV genome contains 12 open reading frames (ORFs): ORF1a, ORF1a’, TF, ORF1b, ORF2a, ORF2b, ORF3, ORF4, ORF5a, and ORF5–ORF7 [12]. Among them, ORF1a and ORF1b encode polyproteins pp1a and pp1b, which can be processed into 17 nonstructural proteins (NSPs) (NSP1α, NSP1β, NSP2, NSP2N, NSP2TF, NSP3-14) that play a major role in virus replication [13,14]. ORF5 and ORF6 encode the major GP5 and M envelope proteins, respectively, which interact with each other to form heterodimers on the surface of virus particles. GP5 is one of the most variable regions of structural proteins in the PRRSV genome and is, therefore, often used for phylogenetic analyses. ORF2, ORF3, and ORF4 encode the minor GP2a, GP3, and GP4 proteins that form noncovalent heterodimers. Two small non-glycosylated proteins E and GP5a are encoded by ORF2b and ORF5a, respectively. ORF7 encodes the highly conserved nucleocapsid protein (N protein) (Figure 1) [15,16,17,18].

GP5 protein is highly variable in PRRSV as it plays an important role in virus infection, as well as simultaneously inducing protective antibodies in the host. Therefore, it is a good target antigen for developing new vaccines [19,20,21]. GP5 protein contains important immune domains related to virus neutralization as well as some peptides or protein motifs, including signal peptides, transmembrane regions, antigenic determinants, and glycosylation sites [22]. As GP5 protein is greatly involved in the invasion, adsorption, and proliferation of PRRSV, it is of great interest to the pathogenicity, diagnosis, prevention, and control of PRRSV [23].

## 2. Overview of GP5 Protein

GP5 protein is a glycosylated envelope protein encoded by the PRRSV ORF5 gene. Its molecular weight is approximately 25 kDa, and it is composed of approximately 200 amino acids [24]. It is divided into four parts: the N-terminal, which is a signal peptide composed of 32 amino acids; the external functional region, which is composed of 35 amino acids containing varying numbers of glycosylated sites; the hydrophobic region, which is composed of 60 amino acids and three transmembrane domains; and the hydrophilic region, which is at the C-terminal end [5]. The extracellular domain of GP5 protein in the American strain contains four N-glycosylation sites, of which N44 and N51 are of the conservative type, which primarily affect the virus infectivity [25]. Glycosylation of GP5 protein may cause the immune escape of the virus, thus reducing the protection of the host’s immune response [26]. The multiple transmembrane hydrophobic regions contained in GP5 protein can make the translated GP5 protein stay in the endoplasmic reticulum, thus inhibiting its expression [27]. GP5 and M proteins of PRRSV are mainly incorporated into virus particles in the form of disulfide-linked heterodimers or polymers, which are essential for virion formation [28].

## 3. Genetic Variation Analysis of GP5 Protein

PRRSV GP5 protein has high genetic variation in the PRRSV genome [29]. GP5 protein of the European strain is composed of 201 amino acids, while that of the American strain is composed of 200 amino acids [30]. The amino acid homology deduced from the PRRSV isotype strain of GP5 is between 88% and 99%, while that deduced from GP5 between the European and American strains is between 52% and 55%. The substitution of deduced amino acids of GP5 among strains of the same type mainly occurs in the hypervariable region (26 aa–39 aa) near the outer region of the signal peptide sequence [31,32], whereas, the amino acid variation of GP5 protein of different strains is mainly concentrated in the signal peptide region, neutralizing epitope region, and non-neutralizing epitope region [33].

Zhang et al. [34] amplified the ORF5 of 16 PRRSV strains in different areas of Shandong Province using reverse transcription-polymerase chain reaction (RT-PCR) and performed a sequence analysis. The results showed that there were site mutations in the amino acids of GP5 protein in all 16 strains, from which 12 strains had mutations at the 29th site of the non-neutralizing epitope and the 34th site of quasi-species evolution, from V to A and N to S, respectively. One strain mutated from A to V at the 185th site of the non-neutralizing epitope. Similarly, Li et al. [11] used RT-PCR to analyze the genetic variation of the isolated PRRSV ORF5 gene. The results showed that the 13th and 151st amino acids and neutralizing epitope (36 aa–52 aa) of ORF5 were mutated, which could lead to changes in virulence and immune evasion. In another study, Lu et al. [35] isolated nine Henan strains and reported that the 39th amino acid of all isolates except HeN-3 had changed from F/L to I (Figure 2). In 2011 and 2012, the mutation of the neutralizing epitope Q40, L41, and the new mutation of the decoy epitope L28 in GP5 protein were found in PRRSV strains in Heilongjiang province [36]. In 2014, the amino acid sites of GP5 in South China were analyzed and found to have extensive variations in the signal peptide region, induced epitope, neutralizing epitope, and hypervariable region of the new branch subgroup [37].

Besides the mutation of amino acid sites, many studies have shown that the 33rd amino acid of PRRSV GP5 protein is deleted. Jiang et al. [38] reported that the GD-HY strain had an amino acid deletion at the 33rd site. Zhou et al. [39] showed that the ORF5 gene of SCcd17 encodes a 199 aa protein and has a novel 1 aa deletion in hypervariable region 1 (HVR1) at the 33rd site. Zhang [40] reported that the 32nd and 33rd amino acids of the strains were deleted, and the glycosylation sites of GP5 were increased and complexed. In addition, H38→K38 mutation occurred in the epidemic strains. Fan et al. [41] reported that SD7, SD8, and SD9 strains were absent in the 34th amino acid. Via phylogenetic analysis based on GP5 amino acids, Sun et al. [42] revealed that the novel NADC30-like PRRSV with a unique single amino acid deletion at the 34th site has become widespread and evolved into a new subgroup.

The gene sequences of representative PRRSV strains were analyzed. A total of 32 target gene sequences of PRRSV GP5 were obtained. To construct the phylogenetic tree, nucleotide sequences of the target gene were aligned using the ClustalX alignment tool and MEGA software (version 7.0.26, Mega Limited, Auckland, New Zealand). Phylogenetic trees were constructed using the neighbor-joining method with 1000 bootstrap replicates in MEGA7.0 (Figure 3). JN-1 and TAXT are similar to the reference strain JXA1, but far from the strain VR2332, and DLS-1 is in a single branch.

Furthermore, Zhao et al. [43] reported that a new insertion site (Lys57) appeared at the 57th site of GP5 protein, which was in the hypervariable region (32 aa–35 aa). PRRSV GP5 protein can easily mutate, which leads to the diversity of strains, increases the difficulty of vaccine development, and makes the disease difficult to control. In conclusion, GP5 protein is highly variable in PRRSV and plays an important role in monitoring variations in PRRSV.

## 4. Immune Function of GP5 Protein

GP5 protein is the main protective antigen targeted by neutralizing antibody induced by PRRSV vaccination or prior infection. The protein has six antigenic determinants. It contains two B cell antigen epitopes: a non-neutralizing epitope A and a neutralizing epitope B. Epitope B induces the body to produce specific neutralizing antibodies against the viral infection. In contrast, epitope A inhibits the recognition of epitope B in the body, thus delaying the production of neutralizing antibodies [30]. Studies show that inserting a Pan-DR T-helper cell epitope (PADRE) between the neutralizing and bait epitopes can reduce or eliminate the bait effect of the non-neutralizing epitope [44]. The reaction between the monoclonal antibody and the expressed product of the deletion mutant shows that GP5 protein has at least two antigenic regions: one extracellular region (27 aa–41 aa) and one at the C-terminal (180 aa–197 aa). The 50 amino acids in the C-terminal of GP5 protein play an important role in maintaining its antigenicity. If these amino acids were absent, GP5 protein would lose its reactivity with the antibody [45].

Akter et al. [46]. showed that the loss of glycan residues in N-linked glycosylation sites (N34, N44, and N51) enhances the immunogenicity of the nearby neutralizing epitope. Leng et al. [47] found that the B antigen region (AR) of HP-PRRSV GP5 did not neutralize AR. The results of indirect enzyme-linked immunosorbent assay (ELISA) and a virus neutralization test showed that there was no correlation between the levels of anti-B AR polypeptide antibody and neutralizing antibodies in the serum of pigs. The specific serum antibody of AR peptide has no neutralizing activity, and glutathione S-transferase-B (GST-B) fusion protein cannot inhibit the neutralizing ability of the antibodies. Three putative N-linked glycosylation sites (N34, N44, and N51) are located in the extracellular domain of GP5, which also include major neutralizing epitopes. Yin et al. [48] indicated that its peptide segments 30–36, 50–55, 140–142, 146–151, and 196–198 may be its dominant B cell epitope regions. Furthermore, T-cell epitopes of PRRSV in GP5 protein have been found (119 aa–127 aa and 151 aa–159 aa) [49]. Zhang et al. [50] used an infectious clone of PRRSV as a vector and replaced Asn with Ala at the 34th or 51st site of GP5 protein. After transfecting the cells, they obtained PRRSV mutants, vVR-N34A and vVR-N51A, which may have been related to the immunogenicity of neutralizing epitope B. The neutralization test results preliminarily suggested that the glycosylation of PRRSV GP5 protein played a role in the host immune function and affected the relationship between the virus and specific neutralizing antibodies. Wang et al. [51] analyzed the carboxyl terminal of PRRSV GP5 protein and predicted and screened the region with a high antigenicity index to construct recombinant phage M13-GP5 168 aa–198 aa. Western blot results showed that the recombinant phage had good reactivity, and the neutralization test confirmed that it could induce piglets to produce high levels of neutralizing antibodies. The latest research shows that a new immune escape mechanism of PRRSV infection has been discovered. Li et al. [52] demonstrated N-terminal acetylation by N-acetyltransferase Nat9 as a novel host defense mechanism that leads to K27-linked-ubiquitination-dependent proteolysis of GP5, which contributes to inhibiting PRRSV infection and proliferation in 3D4/21 cells.

## 5. Apoptosis Induction by GP5 Protein

GP5 protein not only plays a role in PRRSV infection, cell binding, and virus adsorption, but also in cell apoptosis. In 1996, studies revealed that PRRSV GP5 protein caused strong cytotoxicity, and the apoptosis induced by GP5 was independent of anti-apoptotic protein B-cell lymphoma-2 (Bcl-2) [53]. This apoptosis activity could not be prevented by using cell lines that permanently express Bcl-2 protein, which indicates that GP5 either induces apoptosis downstream of Bcl-2 or employs another unknown apoptosis pathway [54]. In vitro, the constructed cells expressing GP5 exhibit typical apoptosis phenomena, such as gradient fragmentation of genomic DNA, reduction of cell number, and formation of apoptotic bodies [55]. Shen [56] found that mutations at different glycosylation sites of PRRSV GP5 protein had different effects on apoptosis induction. Among them, the mutation of N34 and N35 glycosylation sites led to a significant increase in cell apoptosis, whereas the mutation of other sites did not cause any significant effects. Fernández et al. [57] used recombinant vaccinia virus to express PRRSV GP5 protein in mammalian cells. They found that the first 119 amino acids constituted a region that induced apoptosis due to the resulting strong cytotoxicity. The C-terminal region did not induce apoptosis. Hela cells stably expressing GP5 did not show evidence of apoptotic cell death and they speculated that the difference might be due to a histidine tag and an antigenic tag placed in the N-terminal of GP5 fusion constructs. GP5 protein of the PRRSV SD16 strain inhibits virus replication and leads to G2/M cell cycle arrest, but it does not induce Marc-145 cell apoptosis [58]. Similarly, Ma et al. [59] demonstrated that none of the PRRSV structural proteins, including GP5, had the potential to cause apoptosis in Marc-145 cells. Rather, NSP2 and NSP4 played causative roles in PRRSV-induced apoptosis in Marc-145 cells. Some studies have also found that the expression of GP5, GP5^∆84–96^, and GP5^∆97–119^ changes the ratio of Bax/Bcl-2 to different degrees but does not cause the apoptosis of Marc-145 cells [60]. Therefore, the definitive role of GP5 in apoptosis induced by PRRSV is still unknown.

## 6. Effects of GP5 Protein on Virus Replication

GP5 protein plays an important role in PRRSV replication. Song et al. [61] found that a stable expression of GP5 in Marc-145 cells promoted virus replication at the early stage of PRRSV infection by downregulating interferon (IFN) expression; in contrast, interfering GP5 expression with specific small interfering RNA (siRNA) inhibited PRRSV replication. Wang et al. [62] found that the expression of GP5 D84–119 inhibited PRRSV replication by upregulating IFN expression, especially IFN-β, and that the second extracellular region of GP5 played a regulatory role in PRRSV replication. Song et al. [63,64] successfully constructed three short hairpin RNA (shRNA) expression plasmids targeting the PRRSV GP5, which confirmed that the shRNA could effectively inhibit PRRSV replication in Marc-145 cells. They also screened two deoxyribozymes targeting the PRRSV GP5, which confirmed that the designed deoxyribozymes could effectively inhibit PRRSV replication in Marc-145 cells. Furthermore, a study by Wei et al. [65] evidenced that all N-linked glycans in GP5 are not required for virus viability in vitro but are essential for virus replication in vivo. In a different study, Niu et al. [66] studied the effect of PRRSV GP5 protein on the phosphorylation of interferon regulatory factor-3 (IRF-3) in the IFN signaling pathway. Sodium dodecyl-sulfate polyacrylamide gel electrophoresis (SDS-PAGE) and Western blot analysis results showed that GP5 protein could inhibit the phosphorylation of IRF-3.

## 7. Interaction between GP5 Protein and Viral Protein

GP5 and M proteins are the main envelope proteins of PRRSV, which make up more than half of the virus proteins. They can form a heterodimer GP5/M linked by disulfide bonds in virus-infected cells, which is essential for protein transport, assembly, and outflow of progeny virions. Its immunogenicity is strong and conservative, and it allows virus budding to occur on the membrane of the exocytosis pathway [67,68]. The GP5/M protein complex of PRRSV adheres to saliva expressed on cells in order to infect them [69]. When PRRSV was used to infect peptidylgycine α-amidating monooxygenase treated with heparinase, the dimer of M and GP5 proteins was conveniently bound to heparin-like receptors on peptidylgycine α-amidating monooxygenase cells [70]. Zhang et al. [71] found that GP5 and M proteins of PRRSV-1 and PRRSV-2 strains were palmitoylated at cysteine, whether they were expressed alone or in PRRSV-infected cells. Viruses lacking one or two acylation sites in M or GP5 could be saved, but their titers were significantly reduced; in addition, the GP5 and M lacking acylation sites had formed a dimer. Ma et al. [72] used enhanced green fluorescent protein (EGFP) and red fluorescent protein (RFP) as tracers and found that when ORF5-EGFP and ORF6-RFP were co-expressed, GP5 protein could be transported from the endoplasmic reticulum to the high matrix, suggesting that the formation of the GP5/M heterodimer may be related to the post-translational modification, transport, and localization of GP5 protein.

## 8. Interaction between GP5 Protein and Host Proteins

Xue et al. [73] found that the outer domain (GP5-ecto-1) of the first GP5 directly interacts with non-muscle myosin heavy chain 9 (MYH9) C-terminal domain protein (PRA), and this interaction triggers PRA and endogenous MYH9 to form a silk assembly. MYH9 plays a pivotal role in PRRSV infection by physically interacting with PRRSV GP5 protein through its C-terminal domain, which makes the host cells susceptible to infection [74]. Du et al. [75] screened two types of host proteins using immunoprecipitation, namely, the mitochondrial inner membrane protein and calpain protein, both of which interact with GP5. These proteins can co-locate with GP5 in cells and are related to cell protein glycosylation, cell growth, proliferation, movement, function, and maintenance, as well as the development and function of the nervous system. GP5 contains a T4L (T4 lysozyme)-like domain that locally digests the peptide glycan layer during infection. T4 Spackle protein (encoded by gene 61.3) plays a role in inhibiting GP5 lysozyme activity [76]. Hicks et al. [77] found that GP5 protein interacts with *Snapin* of the Marc-145 cell line to take advantage of its role in intracellular transport and membrane fusion. Recent studies revealed that glyceraldehyde 3-phosphate dehydrogenase (GAPDH) interacts with GP5 by binding to 13 of its amino acid sequences (93 aa–105 aa), while GP5 interacts with GAPDH at the K277 amino acid. This indicates that during PRRSV infection, GP5 interacts with GAPDH in the cytoplasm to restrict it from entering the nucleus, and PRRSV promotes virus replication by utilizing glycolysis activity of GAPDH [24]. Zhang et al. In [78] cloned porcine ATP synthase subunit alpha (ATP5A) into the vector pFLAG-CMV-2 and transfected human embryonic kidney (HEK) 293 cells with pCI-GP5 along with pFLAG-CMV-2 or pFLAG-pig-ATP5A. Co-immunoprecipitation was performed using anti-FLAG affinity beads, and the immune complexes resolved by SDS-PAGE were probed with either anti-FLAG or anti-GP5 antibodies. The results showed that ATP5A was readily detected only in the presence of GP5 and not in the presence of empty vector. Thus, it is confirmed that GP5 can interact with ATP5A. Understanding the molecular mechanism of the interaction between PRRSV GP5 and host proteins (Figure 4) can lay the foundation for finding new antiviral targets and exploring the mechanism of viral replication.

## 9. Influence of GP5 on Virulence

Wesley et al. [79] compared the amino acid sequences of VR-2332 with its attenuated vaccine. Their study revealed that the mutation of Arg to Asn at the 13th site and Arg to Gly at the 151st site of the American strain was related to the virulence weakening of the vaccine strain. Hence, the 13th amino acid (R13) and the 151st amino acid (R151) in GP5 protein were the sites related to PRRSV virulence. Studies have also shown that the glycosylation sites of certain Chinese strains have changed remarkably since 2006, and the absence of the 33rd glycosylation site and appearance of the 34th and 35th glycosylation sites may be related to the enhancement of PRRSV virulence [33]. Wang et al. [80] found that the 55th site mutations occurred in GP5 of the Guizhou epidemic strain through cloning and sequencing, and the site mutations at the 25th, 26th, 358th, 409th, and 554th nucleotides could lead to the loss or appearance of some enzyme cut-sites. The 55th site mutations in *GP5* may play a vital role in the enhancement of PRRSV virulence. Most strains have high arginine-glutamine (RQ) motifs near their carboxyl terminal. Further, the highly conserved RQ motifs overlap with the hypervariable GP5 glycosylation sites, which may influence PRRSV virulence [81].

## 10. Applications of GP5 Protein in Vaccines

At present, the PRRSV vaccines used in China are mainly inactivated vaccines, such as CH-1a, or attenuated vaccines, such as CH-1R. Some vaccines on the market are listed in Table 1. The inactivated vaccines have poor immune efficacy, short immune protection time, and limited protection against heterologous strains; whereas the attenuated vaccines pose a risk of virulence reversal and have short immune protection time that cannot prevent a strong viral infection. Therefore, the development of efficient, safe, and affordable vaccines has become a trending research topic in recent years. GP5 protein has good immunological properties and induces neutralizing antibodies, making it the preferred protein for developing new vaccines. These vaccines primarily include nucleic acid, subunit, and live vector vaccines.

Nucleic acid vaccines, also known as DNA vaccines, induce cellular and humoral immunity simultaneously and have cross-protection to different serotypes of strains. Baroed et al. [82] cloned all the ORFs of PRRSV Danish isolates (DK-111/92) in a DNA vaccine vector, and inoculated pigs. The results showed that the pigs immunized with vaccines prepared using ORF1 and ORF4 quickly developed antibody immune responses against NSP2 and GP4; moreover, neutralizing antibody was detected in all pigs, but those inoculated with ORF5 showed the highest antibody titer. Using a DNA-prime/VACV boost regimen, Cui et al. [83] confirmed that the GP5-Mosaic vaccines conferred protection in pigs against heterologous viruses. Vaccination with the GP5-Mosaic-based vaccines resulted in cellular reactivity and higher levels of neutralizing antibodies to both VR2332 and MN184C PRRSV strains. In contrast, vaccination of animals with the GP5-WT vaccines induced responses only to VR2332. GP5-Mosaic vaccine can induce cross-reactive cellular responses to diverse strains, neutralizing antibodies, and protection in pigs [84]. Jiang et al. [85] obtained the suicide DNA vaccine pSFV-56 co-expressed with ORF5 and ORF6 and reported that it had good immunogenicity and could induce immune animals to produce a greater immune response. Jiang et al. [86] co-expressed GP5 and M proteins of PRRSV and found that the resulting dimer significantly improved DNA immunity. Furthermore, cytokines also improve the body’s immune response levels. Qin et al. [87] developed a nucleic acid vaccine PVIR-IL-18-ORF5 containing porcine interleukin-18 (IL-18) and conducted immune experiments in mice. The percentage of CD4- and CD8-positive cells in the pVIR-IL-18-ORF5 experimental group was significantly higher than that in the pVAX1-ORF5 group, which indicated that IL-18 improved both cellular and humoral immunity in animals, with notable improvement in cellular immunity. Li et al. [88] developed a recombinant plasmid carrying the PRRSV GP5 gene (pVAX-GP5) and porcine *interleukin-15* gene (pVAX-IL-15). They inoculated mice with one or both genes and evaluated their humoral and cellular immunity. The proliferation tests showed that the T lymphocyte counts increased in the peripheral blood and spleen of pVAX-GP5-treated mice and were significantly enhanced in combination therapy with pVAX-IL-15. In addition, some studies revealed that adding porcine glutathione peroxidase-1 (GPX1) to PRRSV DNA vaccines produces an adjuvant effect and enhances the humoral and cellular immunity in animals [89]. Recent studies indicate that vaccination with GP5 chimeric DNA vaccine induces cell reaction and high levels of neutralizing antibodies [83]. Although DNA vaccine developments have made great progress, their immune effects are unstable, which warrants further research and improvement.

Recent developments in subunit vaccines have attracted much attention. Plana et al. [90] immunized pregnant sows with insect cell expression products of PRRSV ORF3 and ORF5 of the Spanish isolate (Olot/91) alone or in combination. The results showed that the ORF3 expression product had a protection rate of 68.4%, whereas that of the ORF5 expression product was 50%. Therefore, GP3 and GP5 subunit vaccines have good immunogenicity and are good candidate genes for recombinant subunit vaccine development. Yuan et al. [91] studied the immune-enhancing effects of Taishan Pinus massoniana pollen polysaccharides (TPPPS) and Freund adjuvant on PRRSV GP5 subunit vaccines. The results showed that the recombinant PRRSV GP5 protein induced a significant immune response in animals, which was significantly enhanced by the TPPPS adjuvant, with the middle dose showing the strongest effect. Furthermore, some studies have investigated the use of transgenic Arabidopsis plants expressing codon-optimized and transmembrane deleted antigenic proteins (GP4D and GP5D) as candidate antigens, which produce a good immune response in pigs. Plant-derived GP4D and GP5D proteins provide an alternative platform for producing effective PRRSV subunit vaccines [92]. Guo et al. [93] successfully constructed 10 GP5/M expression vectors. Among the 10 tags, the soluble expression effect of maltose-binding protein (MBP) fused with GP5/M protein was the greatest, and a high-purity recombinant protein of MBP-GP5/M was obtained that laid the foundation for follow-up research and mass preparation of PRRSV subunit vaccines.

Live vector vaccines express the main immunogenic genes of PRRSV through live viruses, such as adenovirus, fowlpox virus, herpes virus, and pseudorabies virus (PRV). Qiu et al. [94] developed recombinant fowlpox virus expressing PRRSV GP5, M, and GP5-M fusion proteins and carried out immunological tests on mice to evaluate its potential in inducing humoral and cellular immune responses. The results showed that, compared with other groups, the recombinant fowlpox virus expressing GP5-M fusion protein stimulated mice to produce high levels of specific neutralizing antibodies, significantly promoted interferon γ (IFN-γ) secretion, and induced the proliferation and expression of specific T lymphocytes, indicating significant enhancement of the humoral and cellular immunity. Wu et al. [95] co-expressed GP5 and M proteins of PRRSV with pseudotyped baculovirus containing hybrid cytomegalovirus (CMV) promoter/SFV (Semliki Forest virus) replicon as vector. The immunogenicity of recombinant baculovirus (BV-SFV-5m6) was compared with that of pseudotype baculovirus vaccine (BV-CMV-5m6), in which the expression of GP5 and M was only driven by CMV promoter. In vitro, BV-SFV-5m6 showed that the expression of foreign protein was enhanced. After immunization in mice, BV-SFV-5m6 induced strong GP5-specific ELISA and neutralizing antibodies against homologous and heterologous viruses. Further, Jiang et al. [96] used live attenuated PRV as the vaccine vector to express GP5 and M proteins in different forms to produce recombinant PRV. The immunized mice subsequently produced PRRSV-specific neutralizing antibodies and had a higher lymphocyte proliferation reaction. Zhao et al. [97] used PRV variant XJ and NADC30-like PRRSV strains (CHSCDJY-2019) as parents to construct a recombinant PRV, rPRV-NC56, with *gE/gI/TK* deletion and co-expression of NADC30-like PRRSV GP5 and M proteins. The results revealed that inoculation of rPRV-NC56 induced specific humoral and cellular immune responses against PRV and NADC30-like PRRSV in mice and protected them from the PRV XJ strain. Furthermore, Cruz et al. [98] expressed GP5 and M proteins of wild-type PRRSV with porcine transmissible gastroenteritis virus as the vector. The findings showed that the vaccinated animals produced faster and stronger humoral and cellular immune responses than unvaccinated animals.

## 11. Applications of GP5 Protein in Detection Methods

At present, there are several detection methods for PRRSV, such as immunoperoxidase monolayer assay to detect PRRS serum antibody, indirect fluorescent antibody assay (IFA), serum neutralization test, and ELISA to detect PRRS antibodies. The immunoperoxidase monolayer assay and IFA are not suitable for large-scale testing as they are subjective, laborious, and expensive; therefore, they are limited to laboratory testing. The serum neutralization test is mainly used to detect neutralizing antibodies produced by PRRSV infection in pigs, but it cannot be used to diagnose acute infectious cases [99,100].

ELISA has strong specificity, high sensitivity, and low costs, so it is widely used in the detection of PRRSV antibodies. PRRSV GP5 protein has good immunogenicity and can induce the body to produce neutralizing antibodies. Li et al. [100] amplified PRRSV *GP5* by RT-PCR, constructed prokaryotic recombinant expression vector (pGEX-6P-GP5), and transferred it into *Escherichia coli* BL21 (DE3) for induced expression. The expressed protein was purified by the gradient urea method and then used as coating antigen. Thus, an indirect ELISA method for detecting PRRSV antibodies was established. Hou et al. [101] deleted the transmembrane region of *GP5* to match the characteristics of the BB0907 strain of HP-PRRSV *GP5* and optimized the remaining gene fragments according to the preference of *Escherichia coli*. Moreover, they constructed the prokaryotic expression vector of the highly expressed GP5 recombinant protein. The indirect ELISA (GP5-ELISA) method for antibody detection of PRRSV GP5 protein was established by optimizing the reaction conditions. Wang [102] prepared the purified recombinant GP5-3m2 protein, used it as the coating antigen, and optimized the indirect ELISA method using the square array test, thereby establishing the PRRSV GP5 indirect ELISA antibody detection method. Li et al. [103] used the gene sequence of NADC30-like PRRSV strain, which is currently widely used as an amplification template in China; comprehensively analyzed the parameters of secondary structural antigen index, antigenicity, hydrophilicity, surface accessibility, and potential neutralizing epitope of structural PRRSV GP5 and M proteins; and selected the corresponding neutralizing epitopes of GP5 and M proteins as the immunogen. By constructing the fusion expression plasmid pET-32a-GP5/M with linker sequence and expressing soluble GP5/M protein in the *Escherichia coli* expression system, an indirect ELISA method for detecting PRRSV neutralizing antibody in pigs was established. All the above mentioned studies show that indirect ELISA possesses advantages of high sensitivity, strong specificity, and good repeatability.

In addition, a new method for PRRSV detection was established. Yang et al. [104] used PCR combined with denaturing high-performance liquid chromatography (DHPLC) to detect PRRSV. Specific primers were designed according to the sequence characteristics of PRRSV GP5, and PCR amplification products were rapidly detected by DHPLC technology. The PCR-DHPLC method established in this study had the advantages of specificity, sensitivity, rapidity, and good repeatability, and could be used for early diagnosis of clinical PRRSV infection and in molecular epidemiological investigations.

## 12. Conclusions

GP5 protein is a highly variable protein in PRRSV, which is the main protective antigenic protein for inducing immunity. GP5 protein promotes virus replication at the early stage of PRRSV infection by downregulating IFN expression; moreover, its N-linked glycan is essential for virus replication in vivo. GP5 and M proteins form the heterodimer GP5-M in virus-infected cells. GP5-M formation may be related to the post-translational modification, transport, and localization of GP5 protein, which is crucial for the assembly, outflow, and virus budding of offspring virions. GP5 protein interacts with PRA and GAPDH in the cytoplasm, thus triggering PRA and endogenous MYH9 to form filament assembly and inhibiting GAPDH from entering the nucleus. In addition, GP5 protein induces apoptosis. For example, GP5 protein expressing PRRSV in mammalian cells using recombinant vaccinia virus provided evidence that the first 119 amino acids form a region that induces apoptosis, whereas the C-terminal region does not. Furthermore, PRRSV detection can be achieved by an indirect ELISA method based on GP5 protein, which possesses high sensitivity, strong specificity, and good repeatability. At present, great progress has been made in the development of new vaccines against PRRSV. However, the immune effect of vaccines is unstable, and the prevention and control of PRRSV are still major concerns that harm the pig industry today. The development of a safer and more effective PRRSV vaccine is needed.

## Figures and Tables

**Figure 1 animals-13-00813-f001:**
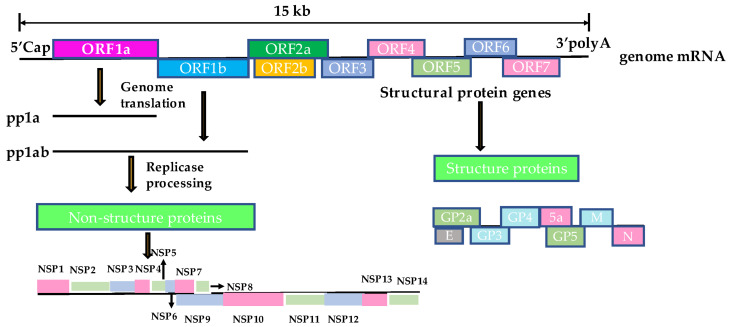
The structural mode of the PRRSV genome.

**Figure 2 animals-13-00813-f002:**
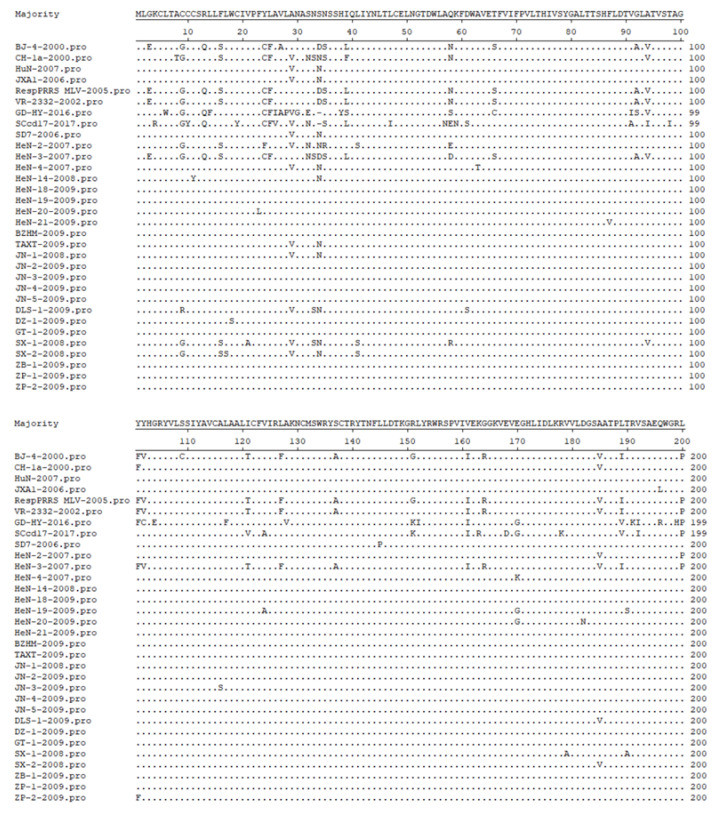
Alignment analysis of amino acid sequence deduced by GP5.

**Figure 3 animals-13-00813-f003:**
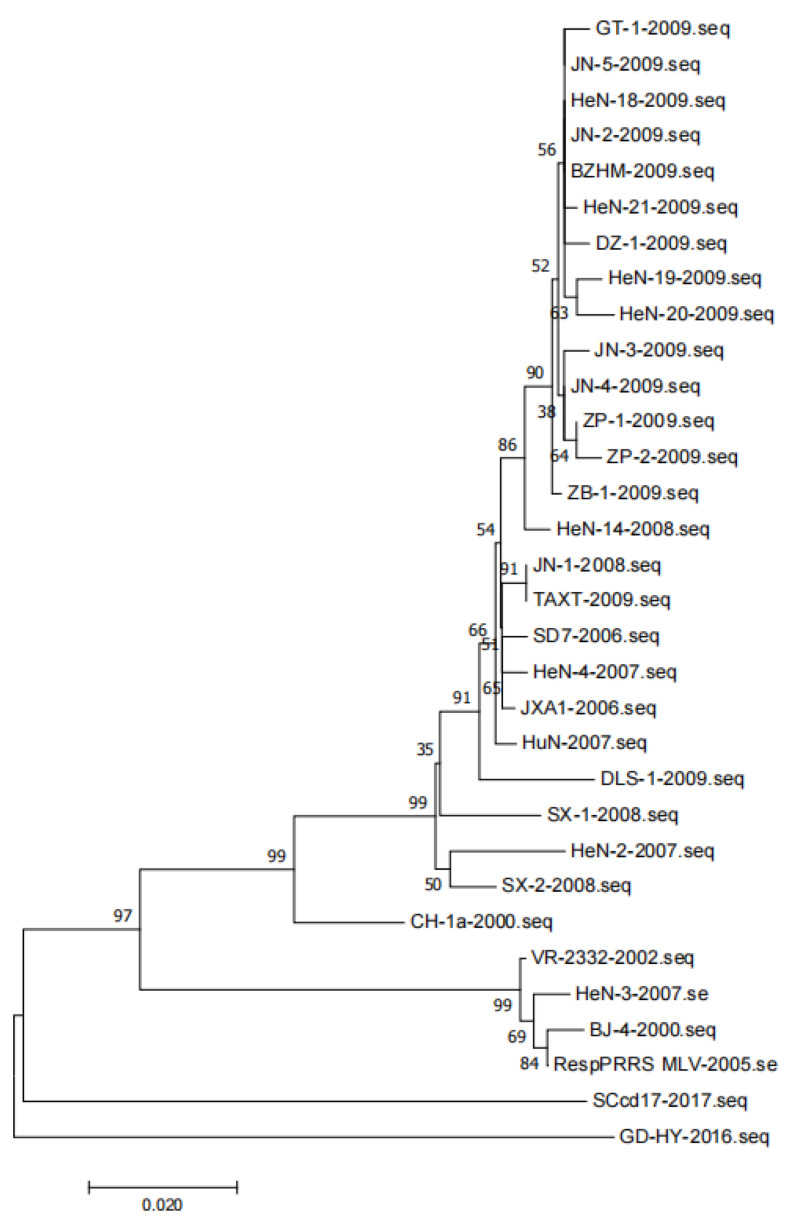
Phylogenetic tree of PRRSV based on the ORF5 gene.

**Figure 4 animals-13-00813-f004:**
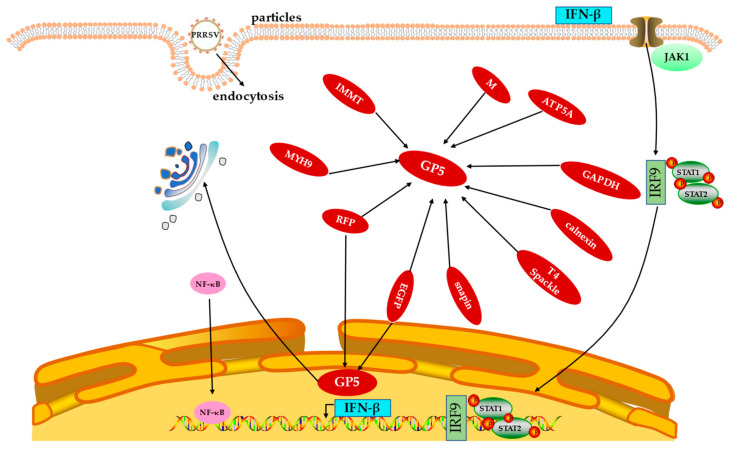
Interactions between PRRSV GP5 protein and host proteins. Abbreviations: GP5, glycoprotein 5; IFN-β, interferon β; PRRSV, porcine reproductive and respiratory syndrome virus; EGFP, enhanced green fluorescent protein; RFP, red fluorescent protein; MYH9, myosin heavy chain 9; GAPDH, glyceraldehyde 3-phosphate dehydrogenase; ATP5A, ATP synthase subunit alpha; IRF9, interferon regulatory factor 9; NF-κB, nuclear factor κB.

**Table 1 animals-13-00813-t001:** PRRSV vaccines on the market.

Year	Area	Vaccine Strain	Vaccine Type
1998	USA	RespPRRS MLV	Attenuated vaccine
1999	USA	MLV RespPRRS/Repro	Attenuated vaccine
2005	China	CH-1a	Inactivated vaccine
2006	USA	Ingelvac ATP	Attenuated vaccine
2006	China	R98	Attenuated vaccine
2006	USA	Prime Pac	Attenuated vaccine
2008	China	CH-1R	Attenuated vaccine
2008	China	TJM-F92	Attenuated vaccine
2011	China	JXA1-R	Attenuated vaccine
2011	China	HuN4-F112	Attenuated vaccine
2018	China	GDr180	Attenuated vaccine
2018	China	PC	Chimeric vaccine

## Data Availability

All datasets are available in the main manuscript. The dataset supporting the conclusions of this article is included within the article.

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
