# Peer review of "Research Progress on Glycoprotein 5 of Porcine Reproductive and Respiratory Syndrome Virus"

_animals, 2023, doi:10.3390/ani13050813_

Round 1
Reviewer 1 Report
The porcine reproductive and respiratory syndrome virus (PRRSV) is one of the most important swine diseases in the world. The authors have experience in studying the PRRS virus, as well as several similar review articles on this pathogen. The article is interesting and worthy of publication; however, some minor issues need to be fixed:
1. Figure 1 is not informative, does not provide additional information or visibility. It may be worth removing this figure, or redoing it to provide more information. For example, mark the pathways in which host proteins take part.
2. It would be interesting if the authors described what the mutations in the amino acids of GP5 protein they noted lead to (lines 101-129).
Reviewer 2 Report
This manuscriopt “
Research progress on glycoprotein 5 of porcine reproductive and respiratory syndrome virus” was described the analysis of various aspects of GP5 protein, in cluding its genetic variation, immune function, interaction with viral protein and host proteins, in duction of cell apoptosis and neutralizing antibodies, influence on virus replication, virulence, and applications in detection methods and vaccines. This information is very important for PRRSV research. The commoms is listed
Major commons
1. The many References was used before 2018, it need more new research studies to support the result described.
2. The many results described is from single research study, is needed more associated researches to support and systemic
3. Section 11 “11. Ability of GP5 to induce neutralizing antibody production”was combined to section 4 “4. Immune function of GP5 protein”
4. Line 153-157: “In a different study, Niu et al. [49] studied 153 the effect of PRRSV GP5 protein on the phosphorylation of interferon regulatory factor-3 154 (IRF-3) in interferon (IFN) signaling pathway. Sodium dodecyl-sulfate polyacrylamide gel 155 electrophoresis (SDS-PAGE) and western blot analysis results showed that GP5 protein 156 could inhibit the phosphorylation of IRF-3.” is removed to section 6 and enhaced it for imparit of PRRSV replication
5. Section 10: please enhance the protection of GP5 vaccine in swine against homologous and heterologous strains.
6. Section 12: It is doubt that the GP5 ELISA is used to diagnosis. It needs to endure the question of gene validation and applicability.
7. The language needs considerable attention. There are so many mistakes that I have not tried to correct them, this needs professional editing by somebody who understands the topic
Mini commons
1. Line 416-418: “Furthermore, PRRSV detection can be 416 achieved by an indirect ELISA method based on GP5--- specificity, and good repeatability.” was delete.
Reviewer 3 Report
I used track changes to make comments and to edit some of the extra words for clarity in select areas.

Reviewer 4 Report
This research summarized the role of GP5 protein of PRRSV in viral infection, replication and immune response, which is beneficial for further study of GP5 protein function.
1. In part “Introduction”: Please add a genome diagram of PRRSV.
2. In part “Genetic variation analysis of GP5 protein”: The authors note that the trend of amino acid variation is not the same for different genotype strains. I suggest the author to use representative strains for evolutionary analysis and amino acid mutation analysis, and the results can be shown in pictures, which is more intuitive.
3. In part “Applications of GP5 protein in vaccines”: Whether a table can be used to summarize vaccines currently on the market or in development.
4. Figure 1: Can you adjust the color? Don't use close colors
Round 2
Reviewer 2 Report
no